# Casein Kinase 1α as a Regulator of Wnt-Driven Cancer

**DOI:** 10.3390/ijms21165940

**Published:** 2020-08-18

**Authors:** Chen Shen, Anmada Nayak, Ricardo A. Melendez, Daniel T. Wynn, Joshua Jackson, Ethan Lee, Yashi Ahmed, David J. Robbins

**Affiliations:** 1Molecular Oncology Program, The DeWitt Daughtry Family Department of Surgery, Miller School of Medicine, University of Miami, Miami, FL 33136, USA; cxs899@miami.edu (C.S.); axn567@med.miami.edu (A.N.); r.melendez@umiami.edu (R.A.M.); DWynn@med.miami.edu (D.T.W.); jjackson@miami.edu (J.J.); 2The Sheila and David Fuente Graduate Program in Cancer Biology, Miller School of Medicine, University of Miami, Miami, FL 33136, USA; 3Department of Cell and Developmental Biology, Vanderbilt University, Nashville, TN 37232, USA; ethan.lee@vanderbilt.edu; 4Department of Molecular and Systems Biology and the Norris Cotton Cancer Center, Geisel School of Medicine, Dartmouth College, Hanover, NH 03755, USA; yashi.ahmed@dartmouth.edu; 5Sylvester Comprehensive Cancer Center, Miller School of Medicine, University of Miami, Miami, FL 33136, USA

**Keywords:** Wnt, cancer, targeted therapies, CK1α, kinase agonists, review

## Abstract

Wnt signaling regulates numerous cellular processes during embryonic development and adult tissue homeostasis. Underscoring this physiological importance, deregulation of the Wnt signaling pathway is associated with many disease states, including cancer. Here, we review pivotal regulatory events in the Wnt signaling pathway that drive cancer growth. We then discuss the roles of the established negative Wnt regulator, casein kinase 1α (CK1α), in Wnt signaling. Although the study of CK1α has been ongoing for several decades, the bulk of such research has focused on how it phosphorylates and regulates its various substrates. We focus here on what is known about the mechanisms controlling CK1α, including its putative regulatory proteins and alternative splicing variants. Finally, we describe the discovery and validation of a family of pharmacological CK1α activators capable of inhibiting Wnt pathway activity. One of the important advantages of CK1α activators, relative to other classes of Wnt inhibitors, is their reduced on-target toxicity, overcoming one of the major impediments to developing a clinically relevant Wnt inhibitor. Therefore, we also discuss mechanisms that regulate CK1α steady-state homeostasis, which may contribute to the deregulation of Wnt pathway activity in cancer and underlie the enhanced therapeutic index of CK1α activators.

## 1. Introduction

The evolutionarily conserved Wnt signaling cascade has been extensively studied for over three decades and has been shown to regulate numerous cellular events during development and adult tissue homeostasis, as well as in disease when deregulated [1,2]. The term ‘Wnt’ was first coined in 1991 from a combination of *wingless*, the gene that patterns the development of many tissues in *Drosophila melanogaster*, including the wing, and its mouse ortholog *Int-1*, the proto-oncogene that regulates mammary tumorigenesis in mice [3]. Wnt ligands are a family of secreted glycoproteins that trigger a unique signal transduction network [4,5]. Wnt proteins undergo palmitoylation by the membrane-bound O-acyltransferase porcupine (PORCN) in the endoplasmic reticulum (ER) [6,7,8]. This modification promotes Wnt export from the ER and subsequently out of the cell and facilitates its activation and binding to the membrane receptor frizzled (Fzd) [6,7,8,9,10]. Two distinct arms of the Wnt signaling network have been identified, defined by their dependence on β-catenin: canonical Wnt/β-catenin signaling and non-canonical Wnt signaling. In this review, we focus on the canonical Wnt signaling pathway, discussing the role of its various critical components in cancer, with a focus on the established negative regulator of Wnt signaling, casein kinase 1α (CK1α).

## 2. Wnt Signaling

### 2.1. At the Membrane

In the absence of Wnt ligands, the transmembrane Wnt receptors Fzd and low-density lipoprotein receptor-related protein 5/6 (LRP5/6) undergo lysosomal degradation, mediated by the E3 ubiquitin ligase zinc- and ring-finger protein 3 (ZNRF3) and its paralog ring-finger protein 43 (RNF43), to maintain low-level Wnt signal transduction (Figure 1) [11,12]. Upon binding of Wnt ligands, Fzd and LRP5/6 oligomerize and recruit disheveled (Dvl) to form the Wnt “signalosome” and initiate signal transduction [13,14,15,16,17,18]. All Wnt ligands activate Dvl, which can stimulate both canonical and non-canonical Wnt signaling, albeit through distinct domains [19]. Additionally, in the presence of Wnt ligands, the R-spondin family of secreted ligands (RSPOs) bind to either leucine-rich repeat-containing G-protein coupled receptors 4/5/6 (LGR4/5/6) or heparan sulfate proteoglycans (HSPG) to attenuate Fzd degradation by ZNRF3/RNF43, thereby increasing the strength of Wnt signaling (Figure 1) [11,12,20,21,22,23,24,25,26,27,28,29].

### 2.2. In the Cytoplasm

In the absence of Wnt, a cytosolic multi-protein complex consisting of the scaffold protein axin, adenomatous polyposis coli (APC), and two protein kinases—casein kinase 1α (CK1α) and glycogen synthase kinase 3 (GSK3)—regulates the phosphorylation and subsequent degradation of the transcriptional regulator β-catenin (Figure 1) [30,31,32,33]. Axin contains multiple functional domains that interact with CK1α, GSK3, and β-catenin and serves to nucleate the formation and stability of this so-called destruction complex. This complex mediates the phosphorylation of β-catenin by CK1α at Ser 45, which primes it for subsequent phosphorylation by GSK3 at Ser 33, Ser 37, and Thr 41 [30,34]. APC also facilitates the phosphorylation of β-catenin in an axin-dependent manner [35,36,37]. Phosphorylated β-catenin is then ubiquitinated by the F-box E3 ubiquitin ligase β-Trcp and subsequently degraded by the proteasome [38,39]. Besides its well-established role in mediating β-catenin’s phosphorylation, APC also attenuates Wnt signaling via a second distinct mechanism [40,41]. In this second role, APC binds to clathrin and adaptor protein 2 (AP2), to prevent LRP6 endocytosis and suppress signalosome formation in the absence of Wnt ligands [40,41].

In response to Wnt ligands, components of the destruction complex are recruited to the plasma membrane, where axin interacts with LRP5/6 to promote signalosome formation and subsequent activation [17,42,43]. In this Wnt on-state, the β-catenin destruction complex is inhibited, resulting in the stabilization and nuclear accumulation of β-catenin (Figure 1) [44,45]. During this process, GSK3 kinase activity is the rate-limiting factor, controlling β-catenin destruction [46,47]. In addition, Wnt signaling can attenuate the formation of the destruction complex through the degradation of axin, which is mediated by the siah E3 ubiquitin ligase 1 (SIAH1) [43,48,49,50].

### 2.3. In the Nucleus

In the absence of nuclear β-catenin, the groucho/transducin-like enhancer of split proteins (Gro/TLE) bind to the T cell factor (TCF)/lymphoid enhancer-binding factor 1 (LEF1) family of transcription factors to repress the Wnt transcriptional program (Figure 1) [51]. Upon Wnt activation, Gro/TLE proteins undergo ubiquitination and degradation, allowing β-catenin to associate with TCF/LEF1 and initiate the Wnt transcriptional program [51,52,53]. This process is modulated by a variety of other co-factors, such as B cell lymphoma 9 protein (BCL9), pygopus (Pygo), and earthbound1 (Ebd1)/jerky [54,55]. Together with these transcriptional co-factors, β-catenin initiates the transcription of a series of Wnt target genes, such as CYCLIN D1, C-MYC, and DICKKOPF1, which play important roles in the regulation of cell proliferation, survival, and differentiation [56,57,58].

## 3. Wnt Signaling and Cancer

The relationship between Wnt signaling and cancer was discovered in the 1980s when the proto-oncogene *Int1* (*Wnt1*) was shown to induce mammary adenocarcinoma in mice [59,60]. A large number of studies have now validated the important role Wnt signaling plays in the onset and progression of a large variety of cancers, largely due to mutations or altered expression of a subset of Wnt signaling components [61,62]. One of the most frequently mutated Wnt signaling components in cancer is *APC*. *APC* is considered the gatekeeper gene of colorectal carcinoma (CRC), as the vast majority of CRCs result from its mutation [61]. Initially, germline mutations of *APC* were discovered to underlie familial adenomatous polyposis (FAP), an inherited precancerous condition that ultimately gives rise to CRC [63,64,65]. Later, somatic *APC* mutations were also shown to play an important role in the development of sporadic CRC [66,67]. *APC* loss-of-function mutations result in increased β-catenin/TCF4-driven transcription, linking the initiation of CRCs to aberrant Wnt pathway activation [31,68,69]. Besides CRC, *APC* mutation is also associated with the genesis of other types of cancers, such as gastric cancer and hepatoblastoma [70,71,72,73]. Other Wnt signaling components are also frequently mutated in cancer, including the genes encoding axin1 [74], β-catenin [75,76,77,78], TCF4 [79,80], and RNF43/ZNRF3 [81,82,83].

Deregulation of the expression of Wnt signaling components is also associated with oncogenesis, both through decreased expression of negative regulators and increased expression of positive regulators [84,85,86,87]. For example, microRNA-135 decreases *APC* expression via the three prime untranslated region (3′-UTR) and leads to Wnt pathway activation in colorectal cancers [88]. Increased protein abundance of Wnt signaling effectors, such as β-catenin and TCF4, due to the deregulation of protein translation, has also been observed in various cancers [89,90,91]. Additionally, the deregulation of the Wnt signaling components’ post-translational modification can also drive tumorigenesis [92,93,94]. For example, in colorectal cancer, TLE3 is ubiquitinated and degraded by RNF6 to facilitate the formation of β-catenin/TCF transcription complex, thereby activating Wnt signaling [94].

Given that constitutive Wnt signaling drives the growth of a large number of cancers, the development of Wnt inhibitors has been a focus of the field for several decades. However, no Wnt inhibitor is currently approved by the U.S. Food and Drug Administration (FDA) for use in the clinic. The major challenge to developing clinically relevant Wnt inhibitors is to overcome the dose-limiting on-target toxicity of such inhibitors in attenuating homeostatic Wnt signaling, primarily in the intestine and bone [2]. We have recently identified several small-molecule CK1α activators that constitute a novel class of Wnt inhibitors capable of attenuating Wnt-driven tumorigenesis without exhibiting significant on-target toxicity [95,96,97,98,99,100,101]. These findings highlight CK1α as a promising therapeutic target in Wnt-driven cancers.

## 4. CK1α

### 4.1. CK1 Family Members

Casein kinases (CKs) were discovered in the 1970s as cytoplasmic protein kinases purified from rat liver, which were able to phosphorylate casein on Ser and Thr residues [102]. Subsequently, multiple CKs were identified and divided into two major groups, CK1 and CK2, based on their biochemical properties [103]. Besides CK1α, the CK1 family of genes encodes CK1β, δ, ε, γ1, γ2, and γ3 [104]. CK1 family members are broadly expressed throughout development and in numerous adult tissues in humans, except CK1β, which is found only in cattle [104]. The primary sequence alignment of CK1 family members highlights a highly conserved Ser/Thr protein kinase domain flanked by distinct amino-terminal (N-term) and carboxyl-terminal (C-term) extensions. Consistent with their homologous protein kinase domain, CK1 family members exhibit similar substrate specificity in vitro. The consensus phosphorylation motif recognized by CK1 was originally identified as a phosphorylated Ser/Thr residue (pSer/Thr) or an acidic group of amino acids upstream of two to four residues, followed by a Ser/Thr phosphor-acceptor [105]. CK1 is also able to phosphorylate substrates at non-consensus sequences [106,107]. For example, CK1α phosphorylates β-catenin at the first serine residue in a novel serine-leucine-serine (SLS) motif upstream of an acidic cluster of six amino acids [107].

### 4.2. CK1 in Wnt Signaling

Despite their substrate similarity in vitro, the substrates of CK1 family members likely vary in vivo. Multiple CK1 family members (CK1α, δ, ε, γ1) regulate Wnt signaling, and this regulation occurs via the phosphorylation of distinct substrates [108]. CK1δ and CK1ε share the highest primary sequence identity and can play a redundant function, such as the phosphorylation and positive regulation of Dvl [109]. CK1ε and CK1γ1 also play positive roles in Wnt signaling, respectively, by phosphorylating TCF3 to enhance its activity [110] or phosphorylating LRP5/6 to enhance signal transduction [111]. In contrast to other CK1 family members, CK1α plays a negative role in Wnt pathway regulation [112,113]. In addition to its well-established role in the cytosolic β-catenin destruction complex, CK1α also regulates the steady-state levels of nuclear Pygo to attenuate β-catenin/TCF-driven Wnt pathway activity [95].

### 4.3. CK1α Splice Variants

The CK1α gene, *CSNK1A1,* undergoes alternative splicing to produce four splice variants [114,115,116]. These splice variants are distinguished by the absence or presence of a long insert (L) of 28 amino acids in the protein kinase domain or a short insert (S) of 12 amino acids near the C-terminus. In human, the four CK1α splice variants include CK1α with both L and S inserts (CK1αLS), CK1α with only an S insert (CK1αS), CK1α with no insert (CK1αNI), and CK1αLS with a truncated N-term (CK1αSN) (Figure 2). The L insert contains a nuclear localization signal (NLS), leading to the preferential nuclear enrichment of CK1α splice variants with this insert [116,117]. CK1α splice variants also exhibit other distinct biological properties, including kinetic characteristics, response to small-molecule modulators, thermal stability, and autophosphorylation [114,115,116,118]. In cells, ectopic expression of the various CK1α splice variants leads to varying phosphorylation of cellular β-catenin on Ser45, suggesting that CK1α splice variants might also affect Wnt pathway activity differentially [119].

### 4.4. Regulation of CK1α

Although the mechanisms by which CK1α regulates cellular processes, including Wnt signaling, are well established, the regulation of CK1α itself is poorly understood and is thus an active area of investigation. Recently, multiple proteins have been described that regulate the intracellular localization of CK1α (Figure 3) [120,121]. For example, in prostate cancers, glioma pathogenesis-related protein 1 (GLIPR1) mediates the translocation of CK1α to the nucleus, leading to the phosphorylation and degradation of C-Myc and inhibition of Wnt activity [120]. The protein levels of CK1α can also be regulated: family with sequence similarity 83G protein (FAM83G) (also known as protein associated with SMAD1 (PAWS1)) interacts with CK1α in the β-catenin destruction complex and stabilizes CK1α protein, subsequently regulating Wnt signal transduction (Figure 3) [122]. In addition, CK1α gene expression and protein abundance are decreased in many Wnt-driven cancers [78,85,97].

The protein kinase activity of CK1α can be regulated by the presence of DEAD-box RNA helicase 3 (DDX3) in the basal state—upon loss of DDX3, CK1α kinase activity is decreased in cells (Figure 3) [123]. However, whether DDX3 directly regulates CK1α is unclear. The P53 inhibitor protein murine double minute X (MDMX) can also inhibit CK1α’s kinase activity upon their binding, resulting in the activation of Wnt signaling (Figure 3) [124]. This result suggests that MDMX, which binds to CK1α in a stoichiometric fashion, functions as a regulatory subunit for CK1α in Wnt signaling. CK1α is also capable of autophosphorylation, which limits its own kinase activity in vitro [118]. Although the autophosphorylation of CK1δ/ε can be reversed by Wnt signaling, there is currently no evidence showing that CK1α autophosphorylation can be regulated by Wnt signaling [125].

## 5. CK1α Activators

### 5.1. Pyrvinium

Based on its important negative role in Wnt signaling, pharmacological activation of CK1α should attenuate Wnt activity. The FDA-approved anthelmintic drug pyrvinium has been the first-in-class small-molecule CK1α activator (Figure 4A), having been identified as a Wnt pathway inhibitor in a large-scale screen of FDA-approved drugs in *Xenopus laevis* embryo extracts [95]. Importantly, pyrvinium has no observable effect on other pathways examined. CK1α is identified as the target of pyrvinium using a candidate approach and then validated in multiple ways. Although it is shown to bind to multiple CK1 family members, pyrvinium only activates the protein kinase activity of CK1α, consistent with pyrvinium acting as a pharmacological CK1α activator [95,119]. Pyrvinium, but not its structural analog VU-WS211 (Figure 4A), activates CK1α by increasing its *V_max_*, without changing its *K_m_* for its substrate [119]. These results suggest that pyrvinium activation of CK1α increases the catalytic activity of CK1α without affecting substrate binding, potentially through an allosteric mechanism (Figure 4B) [95,119]. Interestingly, when comparing the activity of cells transfected with plasmids expressing the four CK1α splice variants, pyrvinium is only able to enhance the activity of those variants lacking the L insert and activated CK1αS the most [119]. Given the location of the L insert within CK1α’s protein kinase domain, which is close to its activation loop (AA 156–190) [126], this result suggests that the L insert may interfere with the binding of pyrvinium to the active site of CK1α.

Consistent with the pivotal role of CK1α in Wnt signaling, pyrvinium attenuates the growth of Wnt-dependent CRC cell lines in a CK1α-dependent manner [95,96]. Pyrvinium also exhibits efficacy against a number of other Wnt-driven cancer cell lines, including those derived from breast cancer and hepatocellular carcinoma [98,99,100,101,127]. Pyrvinium inhibits Wnt pathway activity by reducing β-catenin levels in the cytoplasm and by increasing the degradation of the β-catenin/TCF coactivator Pygo in the nucleus [95]. Consistent with Pygo being a relevant nuclear target of CK1α in Wnt signaling, pyrvinium is able to attenuate the growth of a CRC cell line, harboring a constitutively active β-catenin oncogenic mutant that lacks the CK1α phospho-acceptor Ser [95,96]. Despite its potent Wnt-inhibiting effect ex vivo, the subsequent evaluation of pyrvinium’s efficacy against the growth of Wnt-driven cancers in vivo has been limited, as pyrvinium has low bioavailability outside of the intestinal tract [128]. However, pyrvinium has been evaluated in a Wnt-dependent FAP-induced colorectal adenoma mouse model, in which it significantly decreases the formation of adenomatous polyps in an on-target manner [96]. Based on this work, pyrvinium has been designated by the FDA as an orphan drug for the treatment of FAP.

### 5.2. SSTC Compounds

A number of potent, chemically novel CK1α activators (SSTC3 and SSTC104- see Figure 4), which have significantly improved bioavailability by comparison with pyrvinium, have been described [97,127]. These second-generation CK1α activators bind to CK1α in a manner that is competitive to pyrvinium, suggesting that they bind to a similar site on CK1α [97]. They also attenuate Wnt activity in a manner that is dependent on CK1α. SSTC3, but not its structural analog SSTC111 (Figure 4A), inhibits the growth of Wnt-driven CRC cell lines and patient-derived CRC organoids ex vivo [97]. Consistent with its improved bioavailability, SSTC3 remains in plasma for 24 h after intraperitoneal injection in mice and is able to penetrate the blood-brain barrier [97,129]. Furthermore, SSTC3 significantly inhibits the growth of primary and metastatic tumors that develop from patient CRCs or a Wnt-driven CRC cell line implanted in mice [97]. Importantly, when used at a dose that is efficacious against CRC growth, SSTC3 exhibits no significant toxicity in mice. Specifically, the structure of normal intestinal tissue, which is one of the sites primarily impeded from on-target toxicity of most Wnt inhibitors, is not disrupted by SSTC3 treatment. It has been proposed that the greater therapeutic index of SSTC3 compared to other Wnt inhibitors is the result of the decreased abundance of CK1α protein in CRC tissue versus normal intestinal tissue (Figure 4C). Thus, these reduced levels of CK1α protein sensitize CRC cells to the enhanced activation of CK1α in response to SSTC3 [97].

## 6. Conclusions

Although remarkable effort has been made in the identification and development of Wnt pathway inhibitors, the on-target toxicity of these drugs, including the disruption of bone and intestine tissues, has limited their clinical utility. However, the use of additional drugs that ameliorate these on-target toxicities has extended the potential of currently available Wnt-inhibiting drugs [130,131,132]. For example, a recent phase I trial result has shown that the co-administration of the PORCN inhibitor ETC-159 along with bone protective treatment in patients harboring solid tumors is safe [132]. We have discussed here that CK1α activators show significant efficacy in treating Wnt-driven cancers, without exhibiting on-target toxicity in normal tissue homeostasis at therapeutic doses. Thus, these SSTC compounds have the potential to effectively target Wnt-driven cancers in the clinic, with minimal on-target toxicity.

As a ubiquitous protein kinase, CK1α plays a role in many other cellular processes in addition to Wnt signaling [133]. For example, CK1α is also an important regulator of Ras and Hedgehog pathways, which are two other major signaling pathways regulating numerous disease states [133]. Thus, the therapeutic use of CK1α activators could be extended to other diseases in addition to Wnt-driven cancers. Indeed, it was previously demonstrated that pyrvinium and SSTC3 could also inhibit the growth of Sonic Hedgehog-driven (Shh-driven) medulloblastoma [129,134]. Even though CK1α can regulate both Shh and Wnt pathways, both CK1α activators attenuate Shh activity without affecting Wnt signaling in these medulloblastoma models [129,134]. This suggests that the primary function of these drugs is not limited to Wnt inhibition and is likely context-dependent. However, it still remains possible that CK1α activators can simultaneously regulate multiple cellular processes in contexts other than what has been discussed in this review.

In addition to Wnt- and Shh-driven cancers, CK1α activators have also shown the ability to treat other diseases, such as RAS-driven cancer and Wnt-driven ischemic injury [135,136]. Thus, it becomes quite important to study the mechanisms that regulate CK1α. The known downstream effects of CK1α regulation, e.g., CK1α translocation, gene expression, protein abundance, and kinase activity, may guide the direction of future CK1α studies. In addition, CK1α activators may function by mimicking certain cellular components that can control CK1α activity, suggesting CK1α activators can serve as important tools to identify potential CK1α regulatory machinery. Taken together, the significant cellular roles of CK1α, along with the promising therapeutic effects of CK1α activators in various diseases, have highlighted the importance of understanding the regulation of CK1α.

## Figures and Tables

**Figure 1 ijms-21-05940-f001:**
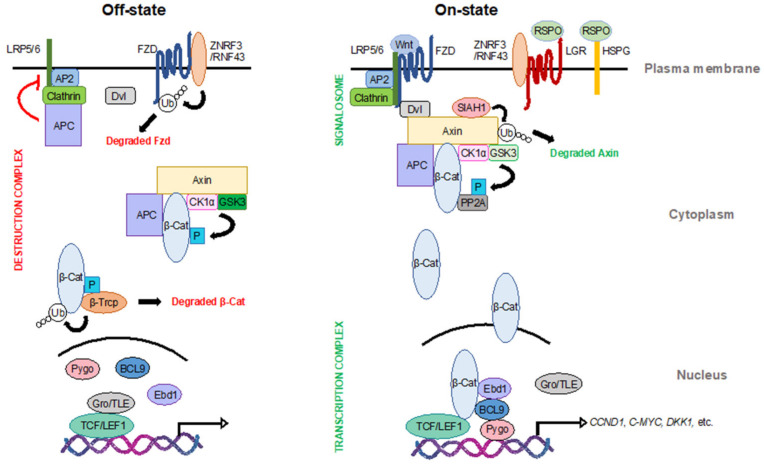
The canonical Wnt signaling pathway. In the Wnt off-state (left), β-catenin, the pivotal transcription coactivator of the Wnt pathway, is degraded by the destruction complex in the cytoplasm. Other Wnt effectors, such as frizzled (Fzd) at the membrane and T cell factor (TCF)/lymphoid enhancer-binding factor 1 (LEF1) transcription factors in the nucleus, are also inhibited to maintain low Wnt activity. In the Wnt on-state (right), Wnt ligands trigger the formation of the signalosome to promote Wnt signal transduction. The function of the destruction complex is inhibited, leading to the stabilization of β-catenin. β-catenin then translocates into the nucleus and binds to TCF/LEF1 to form a transcription complex along with other cofactors to initiate Wnt target transcription. LRP5/6: low-density lipoprotein receptor-related protein 5/6; ZNRF3: E3 ubiquitin ligase zinc- and ring-finger protein 3; RNF43: ring-finger protein 43; Dvl: disheveled; AP2: adapter protein 2; APC: adenomatous polyposis coli; CK1α: casein kinase 1α; GSK3: glycogen synthase kinase 3; Gro/TLE: groucho/transducin-like enhancer of split proteins; Pygo: pygopus; BCL9: B cell lymphoma 9 protein; Ebd1: earthbound 1; RSPO: R-spondin family of secreted ligands; LGR: leucine-rich repeat-containing G-protein coupled receptors; HSPG: heparan sulfate proteoglycans; SIAH1: siah E3 ubiquitin ligase 1; PP2A: protein phosphatase 2A.

**Figure 2 ijms-21-05940-f002:**
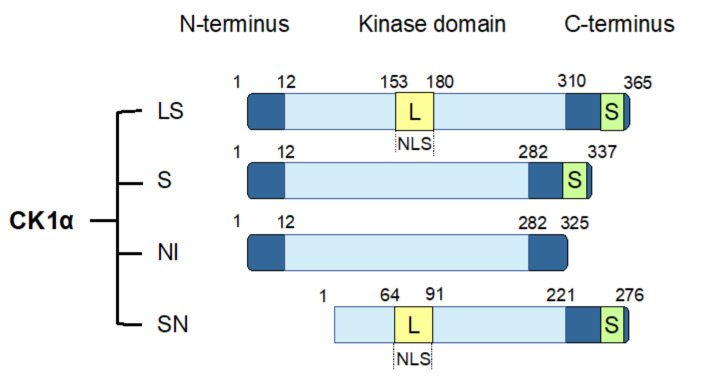
CK1α splice variants. Human CK1α undergoes alternative splicing to produce four splice variants, as shown. These CK1α splice variants are characterized by the insertion of two polypeptide sequences: a long insertion (L) that contains a nuclear localization signal (NLS) into the protein kinase domain, and a short insertion (S) close to the C-terminus. LS: CK1α with both L and S inserts; S: CK1α with only an S insert; NI: CK1α with no insert; SN: CK1α LS with an N-terminal truncation.

**Figure 3 ijms-21-05940-f003:**
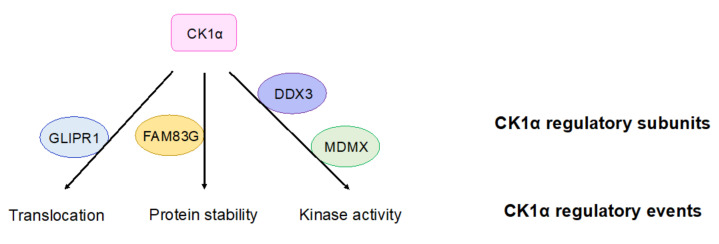
CK1α regulatory subunits. Proteins that have been reported to regulate CK1α are shown. These proteins bind to CK1α and lead to indicated regulatory outcomes of CK1α. GLIPR1: glioma pathogenesis-related protein 1; FAM83G: family with sequence similarity 83G protein; DDX3: DEAD-box RNA helicase 3; MDMX: murine double minute X.

**Figure 4 ijms-21-05940-f004:**
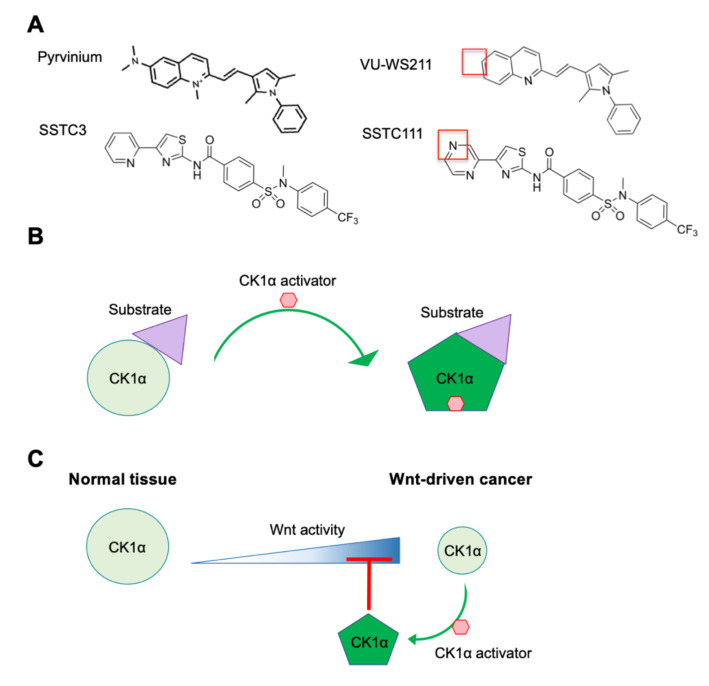
CK1α activators. (**A**) The structures of two chemically distinct CK1α activators—pyrvinium and SSTC3—and their inactive analogs—VU-WS211 and SSTC111—respectively, are shown. The red boxes highlight key structures needed for maximal efficacy. (**B**) A model, highlighting how CK1α activators function to increase the catalytic efficiency of CK1α. (**C**) A model of the mechanism underlying the differential therapeutic index of CK1α activators in normal tissue and Wnt-driven cancer.

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
