# Peer review of "Casein Kinase 1α as a Regulator of Wnt-Driven Cancer"

_ijms, 2020, doi:10.3390/ijms21165940_

Round 1
Reviewer 1 Report
Thank you for allowing me to review the manuscript entitled “Casein Kinase 1a as a regulator of Wnt-driven cancer.” Overall, the review article is well-written and efficiently outlines the role of CK1a and its interactions in the Wnt pathway. The manuscript may benefit from select modifications, but is overall appropriate for publication.
The Wnt pathway is increasingly recognized as a component of oncogenesis via deregulation of the pathway. Investigations into influencing this pathway are in progress, and potential actionable sites would be of therapeutic value. The authors postulate CK1a as such an agent and provide a review of the kinase and its properties as currently understood. This facilitates understanding of its interactions with the Wnt pathway as well as potential avenues for therapeutic intervention via novel drug agents.
General comments: The authors have efficiently outlined their contextualizations of characteristics of CK1a with attention to potential clinical relevance. In particular, the section on pyrvinium was succinct and applicable.
Some questions, although minor, include:
- Lines 254-259: consider including some more detail on methodology of these studies and their findings. SSTC3 appears to be a significant potential target and expansion of this section would better inform potential researchers. This would enhance the applicability of this review.
- Addition of a “future directions” or ongoing studies section would also increase the utility of this review.
- In the conclusion section, there is mention of the involvement of CK1a in other pathways. May benefit the overall scope of the article to detail this a bit further to better contextualize the importance of further investigation into CK1a.
- RSPOs were briefly discussed in the section on Wnt signaling at the membrane, then never addressed again or related to the larger article. Consider better integration or elimination if not salient.
Overall, this is a well-written manuscript with concise and appropriate overview of Ck1a interactions in the Wnt pathway. It contributes a useful overview to the existing literature and may facilitate better understanding of these interactions, enabling further research. After addressing the above points, this would be an appropriate article for publication.
Author Response
1. We thank the reviewer for the insightful comments. We have revised the text as suggested:
Line 250-256 in the revised manuscript: “Consistent with its improved bioavailability, SSTC3 remained in plasma for 24 hours after intraperitoneal injection in mice and was able to penetrate the blood-brain barrier [97,129]. Furthermore, SSTC3 significantly inhibited the growth of primary and metastatic tumors that developed from patient CRCs or a Wnt-driven CRC cell line implanted in mice [97]. Importantly, when used at a dose that was efficacious against CRC growth, SSTC3 exhibited no significant toxicity in mice. Specifically, the structure of normal intestinal tissue, which is one of the sites primarily impeded from on-target toxicity of most Wnt inhibitors, was not disrupted by SSTC3 treatment.”
2. As suggested, we have added a few sentences of future directions :
Line 293-299 in the revised manuscript: “In addition to Wnt- and Shh-driven cancers, CK1a activators have also shown ability to treat other diseases such as RAS-driven cancer and Wnt-driven ischemic injury [135,136]. Thus, it becomes quite important to study the mechanisms that regulate CK1a. The known downstream effects of CK1a regulation, e.g. CK1a translocation, gene expression, protein abundance, and kinase activity, may guide the direction of future CK1a studies. In addition, CK1a activators may function by mimicking certain cellular components that can control CK1a activity, suggesting CK1a activators can serve as important tools to identify potential CK1a regulatory machineries.”
Additionally, we have previously addressed the potential of DDX family proteins and autophosphorylation as CK1a regulatory mechanisms (see Line 202, 206-208 in the revised manuscript).
3. As suggested, we have added a few sentences to contextualize the study of CK1a and its activators:
Line 281-282 in the revised manuscript: “…CK1a plays a role in many other cellular processes in addition to Wnt signaling [133]. For example, CK1a is also an important regulator of RAS and Hedgehog pathways, which are two other major signaling pathways regulating numerous disease states [133]. Thus, the therapeutic use of CK1a activators…”
Line 286-291 in the revised manuscript: “Even though CK1a can regulate both Shh and Wnt pathways, both CK1a activators attenuate Shh activity without affecting Wnt signaling in these medulloblastoma models [129,134]. This suggests that the primary function of these drugs is not limited to Wnt inhibition and is likely context-dependent. However, it still remains possible that CK1a activators can simultaneously regulate multiple cellular processes in contexts other than what have been discussed in this review.”
4. As suggested, we have integrated the RSPO section with Wnt ligands (see Line 67-70 in the revised manuscript).
Reviewer 2 Report
The authors wrote an exhaustive review about the Casein Kinase 1α modulator of Wnt pathway, one of the major oncogene driver in cancer. The authors then focused on CK1α activators as new therapeutic option to overcome the toxicity of Wnt inhbitors.
The review is clear, straightforward and well written. No modification are required in this reviewer's opinion.
I would only add a sentence regarding the ability of compound SSTC3 to cross the BBB, which is impressing and useful for Wnt-driven brain tumors.
Author Response
We thank the reviewer for the insightful comments. We have added the ability of SSTC3 to cross the BBB as suggested:
Line 250-252 in the revised manuscript: “Consistent with its improved bioavailability, SSTC3 remained in plasma for 24 hours after intraperitoneal injection in mice and was able to penetrate the blood-brain barrier [97,129].”
Reviewer 3 Report
This is an excellent review co-authored by renowned experts in the field. It gives a perfect overview of the Wnt pathway in general and highlights the role of casein kinases 1 (CK1) in Wnt signalling, specifically focusing on recently identified activators of CK1a for the inhibition of the pathway in cancer. I suggest to publish the review as it stands after eliminating few typographical errors (e.g. ref 119 in list).
Author Response
We thank the reviewer for the insightful comments. We have corrected all typos that we notice.
Reviewer 4 Report
The review article, "Casein Kinase 1a as a regulator of Wnt-driven cancer" was a pleasure to read. The authors identified a key niche that has not been extensively covered in the literature and provided a very succinct introduction to CK1 and the drugs being used to target it. One issue that occured as I read the piece was whether the authors anticipate non-wnt specific effects of using CK1 drugs.
There were a few minor typos.
Author Response
We thank the reviewer for the insightful comments. We have corrected the typos that we notice.
We have also added the finding that pyrvinium does not affect other signaling pathways including BMP4, IL4, TGF-a, and Notch pathways while inhibiting Wnt signaling:
Line 217-218 in the revised manuscript: “Importantly, pyrvinium has no observable effect on other pathways examined.”
Additionally, we have included the context dependency of these drugs:
Line 286-291 in the revised manuscript: “Even though CK1a can regulate both Shh and Wnt pathways, both CK1a activators attenuate Shh activity without affecting Wnt signaling in these medulloblastoma models [129,134]. This suggests that the primary function of these drugs is not limited to Wnt inhibition and is likely context-dependent. However, it still remains possible that CK1a activators can simultaneously regulate multiple cellular processes in contexts other than what have been discussed in this review.”